# Identification of small molecule agonists of fetal hemoglobin expression for the treatment of sickle cell disease

Jian-Ping Yang[1], Rachel Toughiri[1], Anshu P. Gounder[1], Dan Scheibe[1], Matt Petrus[1], Sarah J. Fink[2], Sebastien Vallee[2], Jon Kenniston[2], Nikolaos Papaioannou[2], Steve Langston[2], Narender R. Gavva[1], Shane R. Horman[1]*

1 Takeda Development Center Americas, Inc., San Diego, California, United States of America, 2 Takeda Development Center Americas, Inc., Cambridge, Massachusetts, United States of America

* shanehorman@gmail.com

**Data Availability Statement:** All relevant data are within the manuscript and its Supporting information Data files.

## Abstract

Induction of fetal hemoglobin (HbF) has been shown to be a viable therapeutic approach to treating sickle cell disease and potentially other β-hemoglobinopathies. To identify targets and target-modulating small molecules that enhance HbF expression, we engineered a human umbilical-derived erythroid progenitor reporter cell line (HUDEP2_HBG1_HiBiT) by genetically tagging a HiBiT peptide to the carboxyl (C)-terminus of the endogenous *HBG1* gene locus, which codes for γ-globin protein, a component of HbF. Employing this reporter cell line, we performed a chemogenomic screen of approximately 5000 compounds annotated with known targets or mechanisms that have achieved clinical stage or approval by the US Food and Drug Administration (FDA). Among them, 10 compounds were confirmed for their ability to induce HbF in the HUDEP2 cell line. These include several known HbF inducers, such as pomalidomide, lenalidomide, decitabine, idoxuridine, and azacytidine, which validate the translational nature of this screening platform. We identified avadomide, autophinib, triciribine, and R574 as novel HbF inducers from these screens. We orthogonally confirmed HbF induction activities of the top hits in both parental HUDEP2 cells as well as in human primary CD34+ hematopoietic stem and progenitor cells (HSPCs). Further, we demonstrated that pomalidomide and avadomide, but not idoxuridine, induced HbF expression through downregulation of several transcriptional repressors such as BCL11A, ZBTB7A, and IKZF1. These studies demonstrate a robust phenotypic screening workflow that can be applied to large-scale small molecule profiling campaigns for the discovery of targets and pathways, as well as novel therapeutics for sickle cell disease and other β-hemoglobinopathies.

## Introduction

Sickle cell disease (SCD) is an inherited blood disorder caused by a single amino acid substitution in the human β-globin protein (HBB) from a hydrophilic glutamine to a hydrophobic

**Funding:** The funder provided support in the form of salaries for authors, but did not have any additional role in the study design, data collection and analysis, decision to publish, or preparation of the manuscript. The specific roles of these authors are articulated in the 'author contributions' section.

**Competing interests:** Takeda Development Center Americas, Inc. provided sponsorship and financial support for this study. All the authors are employees of Takeda Pharmaceutical Industries, Ltd., and had equity ownership with Takeda Pharmaceutical Industries, Ltd. The Takeda commercial affiliation does not alter our adherence to PLOS ONE policies on sharing data and materials. There are no patents, products in development or marketed products associated with this research to declare.

valine (Glu6Val) [1]. It is estimated that 100,000 people are affected in the US, and millions worldwide [2, 3]. Fetal hemoglobin (HbF), expressed in the early gestational stage, is a form of hemoglobin comprised of 2 α chains and 2 γ chains [4]. Soon after birth, γ-globin is replaced with β-globin through a process called globin-switching, yielding adult hemoglobin (HbA) [5, 6]. In SCD patients, the mutated β-globin protein causes hemoglobin polymerization in red blood cells, resulting in various complications such as hemolysis, vaso-occlusive crisis, vasculo-pathy, and subsequent inflammation and end-stage organ damage [7, 8]. Despite the severity of the disease and these symptoms, SCD patients have few treatment options to date [9].

Preclinical studies as well as patient-derived data indicate that restoring expression of HbF can ameliorate severity of the disease; this is considered to be an important therapeutic strategy for both β-thalassemia and SCD [10–13]. Currently, hydroxyurea (HU) is the only FDA-approved therapeutic treatment for SCD capable of increasing HbF levels in SCD patients [14]. Although HU treatment may reduce morbidity and mortality in adults and children suffering from SCD, its therapeutic effect is limited in many patients; HU often demonstrates adverse effects and potential toxicities, particularly for long-term treatment [15, 16]. Several anti-neoplastic drugs such as sodium butyrate [12], 5-azacitidine [17], and decitabine [18] have been explored as HbF inducers. However, clinical usage of these pharmacological agents is limited due to unfavorable side effect profiles [9, 19]. Therefore, novel small molecule approaches to induce HbF therapeutically with limited toxicity and favorable economic accessibility are urgently needed [20].

The systematic profiling of chemical agents to identify those capable of reactivating γ-globin expression has been hampered by the lack of suitable erythroid cell lines and methods. Recent reports of small molecule screens for fetal hemoglobin inducers have mainly employed reporter cell lines engineered with a partial γ-globin promoter driving a reporter gene, such as green fluorescent protein (GFP) or luciferase [21–24]. Although tractable for high-throughput compound screening, these synthetic gene constructs may not address essential endogenous DNA elements that tightly regulate and control γ-globin gene expression in erythroid cells. Further, until recently, many cell-based screens for HbF-inducers have been conducted in human immortalized myelogenous leukemia K562 cells [25, 26] or murine erythroleukemia MEL cells [27], which are poor surrogates for normal erythroid cells and thus are not optimal systems for studying hemoglobin switching and regulation. In addition, the use of flow cytometry assays (FACS) or ELISA-based readouts for HbF quantitation have constrained their widespread application due to time-consuming and expensive assay protocols that are poorly adaptable to large-scale industrial drug discovery screening campaigns.

To address the limitations of conventional HbF screening platforms and to accelerate the pace of sickle cell disease therapeutic discovery, we employed the clustered regularly interspaced palindromic repeat (CRISPR)-Cas9 nuclease system to generate a human umbilical-derived erythroid progenitor (HUDEP2) γ-globin reporter cell line by inserting a HiBiT tag to the C-terminal portion of the endogenous *HBG1* gene. HUDEP2 cells are an immortalized human CD34+ hematopoietic stem cell line and can differentiate into mature (α2β2 type) erythrocytes [28]. HUDEP2 cells normally express very low levels of γ-globin upon erythroid differentiation, providing an advantage when using this cell line for screening compounds that restore endogenous *HBG1* expression. This genetically engineered erythroid reporter cell line enables real time interrogation of endogenous expression levels of γ-globin via a simple luminescence readout. As a proof-of-concept study, we miniaturized the assay to accommodate a 384-well plate format and performed a high-throughput chemogenomic phenotypic screen using an annotated small molecule library. The hits from this screen can be used to explore potential targets and pathways for novel HbF inducers, as well as the potential for clinical repositioning of existing drugs for treatment of SCD. The screening platform was able to 1) identify

internal positive control compounds and their derivatives and 2) reveal novel molecules/compounds with HbF-inducing properties. Moreover, induction of HbF by hit compounds in human primary CD34+ hematopoietic stem cells confirmed the validity of our screening assay.

## Materials and methods

### HUDEP2 cell culture

A two-phase liquid culture system was used to culture HUDEP2 cells (obtained from the RIKEN Institute) at 37˚C and 5% $CO_2$ in a humidified incubator. The Phase I expansion media contains 50 ng/mL human stem cell factor (R&D Systems), 10 µM dexamethasone (Sigma-Aldrich), and 1 µg/mL doxycycline (Sigma-Aldrich) in StemSpan SFEM media (Veritas USA). The Phase II differentiation medium comprises Iscove's Modified Dulbecco's Medium (StemCell Technologies), 2% fetal bovine serum (Thermo Fisher Scientific), 3 U/mL erythropoietin (R&D Systems USA), 10 µg/mL insulin (StemCell Technologies), 500 µg/mL holo-transferrin (Sigma-Aldrich), and 3 U/mL heparin (StemCell Technologies) for differentiating progenitor cells into erythrocytes.

### *Gene knock-in* using CRISPR/gRNA RNP electroporation

The 3'-phosphorothioate chemically modified single CRISPR gRNA and single strand nucleotide of knock in template were synthesized by Integrated DNA Technologies (IDT). The sequence of single-stranded nucleotide is ATCTCTCAGCAGAATAGATTTATTATTTGTATTG CTTGCAGAATAAAGCCTATCCTTGAAAGCTCTGAATCATGCCCAGTGAGCTCAGCTAAT CTTCTTGAACAGCCGCCAGCCGCTCACGGAGACGTGGTATCTGGAGGACAGGGCACTGGC CACTGCAGTCACCATCTTCTGCCAGGAAGCCTGCACCTCAGGG. The SpCas9 NGG protospacer adjacent motif (PAM) sequence was mutated from "GGG" to "GCC"" in the single-stranded oligodeoxynucleotide (ssODN) donor template to avoid repeated cutting after the tag insertion. The gRNA1 sequence targeting the *HBG1* gene is: CCTTGAAAGCTCTGAATCAT. TE buffer was used to resuspend lyophilized sgRNA from Integrated DNA Technologies. The sgRNA was mixed with TrueCut Cas9 v2 protein (Thermo Fisher Scientific) and incubated for 10 minutes to generate RNP complex. After cell counting, 100,000 HUDEP2 cells were transfected using Neon transfection system (Thermo Fisher Scientific) and then transferred to a 96-well plate with 100 µl of prewarmed expansion media for 48 hours. The single cell cloning was done by limited dilution and seeding cells at a density 0.5 cells/well in 100 µl of expansion media in the 96-well plates. The single cell clones were then screened for insertion by PCR using PCR primers covering the insert fragment.

### Single cell cloning and confirmation

Bulk transfected HUDEP2 cells were incubated in expansion media for 48 h and were then plated clonally at limiting dilution in 96-well plates. After approximately 20 days of clonal expansion, the cells in each well were split into two. The cells in one set were harvested to extract the genomic DNA with 50 µL of QuickExtract DNA Extraction Solution per well (Epicentre), which were subsequently used for PCR screening using Phusion High-Fidelity PCR Master Mix (Thermo Fisher Scientific). PCR positive clones were then expanded and tested for their response to pomalidomide treatment (positive control compound) by Nano-Glo® HiBiT Lytic Detection assay (Promega). The PCR products of 4 positive clones that responded to pomalidomide treatment were subjected to next generation sequencing to confirm the insertion of the HiBiT tag at the appropriate position of the *HBG1* gene.

## Chemical library

The SelleckChem compound library (catalog numbers: L3800 and L1100) contains approximately 5000 compounds annotated with known/predicted targets or mechanisms of action (MoA) that have either entered clinical studies or been approved by the FDA. Each compound was dissolved in dimethylsulfoxide (DMSO) at a concentration of 10 mM.

## High-throughput screening using HUDEP2_HBG1_HiBiT

Compounds at a stock concentration of 10 mM in DMSO were spotted into the wells of 384-well plates by automated acoustic liquid dispensing. HUDEP2_HBG1_HiBiT cells were harvested from HUDEP2 expansion media and suspended in differentiation media at a concentration of $2x10^5$ cells/ml. 50 μl of cells was added to each well of the 384-well plates using Multidrop™ Combi Reagent Dispenser (Thermo Fisher Scientific) to yield final concentrations of either 10 μM or 1 μM. After 5-days of incubation, 20 μl of Nano-Glo® HiBiT Lytic Detection reagent (Promega) was added to wells. Luminescence signals were measured according to the manufacturer's protocol using an EnVision Microplate Reader (PerkimElmer). Screening data were analyzed using TIBCO Spotfire software (PerkimElmer).

## Validation of hits in HUDEP2_HBG1_HiBiT cells

HUDEP2_HBG1_HiBiT cells ($2x10^5$ cells/well/50 μl) in 384-well plates were incubated with serially diluted compounds in differentiation medium at 37°C for 5 days. For HiBiT luminescence assays, 20 μl of Nano-Glo HiBiT Lytic Detection reagent (Promega) was added. For cell viability assays, 20 μl of CellTiter-Glo 2 reagent was added to each well according to the manufacturer's protocol (Promega).

## Human primary CD34+ cell culture

Human primary CD34+ HSPCs from G-CSF-mobilized healthy adult donors were obtained from StemCell Technologies. CD34+ HSPCs were expanded, then subjected to two phase liquid culture for erythroid differentiation. Briefly, HSPCs were thawed on day 0 into CD34 + expansion media consisting of Stem Span SFEMII supplemented with 1x StemSpan™ CD34 + expansion supplement and 1 μM of U729 (StemCell Technologies). On day 7, the cells were switched into Phase I erythroid expansion media consisting of Stem Span SFEMII supplemented with 1x erythroid expansion supplement (StemCell Technologies), 1 pM of dexamethasone (Sigma-Aldrich) and 1% penicillin/streptomycin (Thermo Fisher Scientific) for 7 days. Subsequently, the media were switched into Phase II erythroid differentiation media consisting of Stem Span SFEMII supplemented with 3% normal human serum (Sigma-Aldrich), 3 IU/ml erythropoietin (R&D Systems) and 1% penicillin/streptomycin. Cells were maintained at a density of $0.1-1x10^6$ cells/ml with media changes every other or every third day as necessary.

For compound treatment, CD34+ cells were first cultured in phase I expansion media in a flask for 4 days, then seeded in 24-well plates at a density of $1.5x10^5$ cells in 300 μl of phase I expansion media supplemented with various concentrations of test compounds for 3 days at 37°C and 5% $CO_2$. Phase II differentiation media and compounds were replenished after 3 days of culture. After 7 days of incubation at 37°C and 5% $CO_2$, cells were harvested for either HbF analysis by FACS or protein analysis by western blot.

## Flow cytometry analysis

For HbF analysis, cells were washed with 1xDPBS and stained with fixable violet dead cell stain dye (Thermo Fisher Scientific), then fixed in 0.05% glutaraldehyde (Sigma-Aldrich) for 10 min at room temperature. Following that, cells were washed with stain buffer (BD Pharmingen) 2 times, and permeabilized with 0.1% Triton X-100 in DPBS (Thermo Fisher Scientific) for 5 min at room temperature. Following one wash with stain buffer, cells were stained with a HbF-APC conjugate antibody (Thermo Fisher Scientific) for 30 minutes in the dark. Cells were then washed twice with stain buffer. Flow cytometry was carried out on an Attune™ Flow Cytometer (Thermo Fisher Scientific).

## Western blot analysis

HUDEP2 cells treated with compounds in differentiation media for 5 days were collected by centrifugation, media aspirated, then washed with DPBS prior to lysis. Cell pellets were lysed using RIPA Lysis and Extraction Buffer (Thermo Fisher Scientific) supplemented with complete Protease Inhibitor Cocktail (Millipore Sigma). Protein concentrations in cleared lysate were quantified by BCA Assay (Pierce). Automated western blotting was performed using the 12–230 kDa Separation Module for PeggySue (Bio-Techne) according to the manufacturer's protocols. 5ul of 0.1–0.2mg/ml of cell lysate was used for analysis of indicated protein targets. The following antibodies were used for immunodetection. Mouse anti-Ctip1/BCL-11A antibody (Abcam), rabbit anti-alpha globin (Abcam), rabbit anti-hemoglobin beta/ba1 (Abcam), rabbit anti-hemoglobin gamma (Cell Signaling Technology), rabbit anti-ZBTB7A/LRF/Pokemon (Cell Signaling Technology), rabbit anti-CRBN (Abcam), rabbit anti-Ikaros/IKZF1(Cell Signaling Technology), rabbit anti-GAPDH (Cell Signaling Technology).

# Results

## Generation of HUDEP2_HBG1_HiBiT reporter cell line

The HiBiT system is a protein complementation assay consisting of a split NanoLuc luciferase enzyme. HiBiT, a short 11 amino acid peptide, binds with high affinity to another larger subunit called LgBiT. The two protein subunits are reconstituted to form a complex of NanoLuc enzyme, which yields a dynamic luciferase signal in the presence of added furimazine substrate [29] (Fig 1A). Since HiBiT is a relatively small tag, developed as a non-invasive way to tag endogenous proteins, the possibility of it affecting normal protein function is unlikely.

To monitor endogenous γ-globin expression, we generated a reporter cell line by knocking in the HiBiT tag to the C-terminus of *HBG1* gene via CRISPR/Cas9-mediated homologous recombination (Fig 1B). The tandem *HBG1* and *HBG2* genes, which code [A]γ-globin and [G]γ-globin respectively, harbor nearly identical nucleotide sequences, making it challenging to specifically tag only one gene. In order to avoid large sequence deletions and to insert the HiBiT tag selectively in the C-terminus of the *HBG1* gene, we designed a gRNA sequence fragment containing a single nucleotide difference in the non-coding regions of the C-termini between the two genes. This gRNA enabled the Cas9 enzyme to selectively cut the *HBG1* gene 17 base pairs from the stop codon (Fig 1C). From 250 single cell clones, we obtained 4 single cell clones that showed a significant positive response to treatment with pomalidomide, an orally active thalidomide analog that has demonstrated the ability to upregulate HbF production *in vitro*, in sickle mice, and in Phase I clinical trials [30–32]. Among these clones, two single cell clones had a heterozygous gene knock-in of the HiBiT fragment at the C-terminus of *HBG1* gene in one allele based on sequence analysis. We chose the one single cell clone (renamed as HUDEP2_HBG1_HiBiT reporter cell line) that had the lowest background and highest signal:

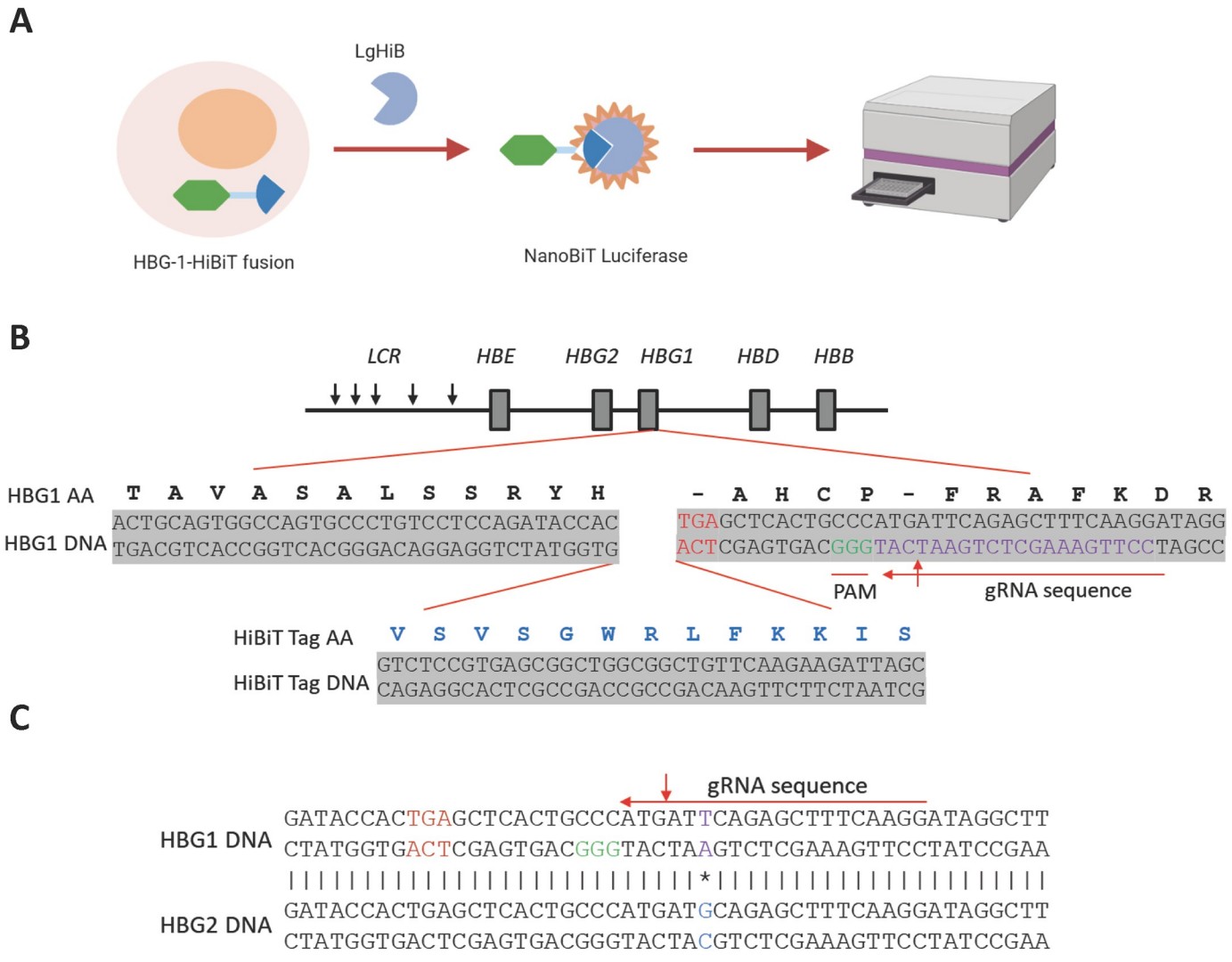

**Fig 1. Engineering of HUDEP2_HBG1_HiBiT reporter cell line.** (A) Schematic representation of HiBiT protein complementation assay. Cells expressing a fusion protein of target and small HiBiT reconstituted with large subunit LgBiT to form a NanoLuc complex, which generates a luminescent signal in the presence of added furimazine substrate. (B) Schematic representation of insertion of HiBiT tag to the C-terminus of the *HBG1* gene within the γ-globin locus. The stop codon of *HBG1* gene is in red. The gRNA sequence and corresponding PAM sequence are indicated. (C) The sequence of gRNA that is designed to selectively target the *HBG1* gene. Alignment of C-terminal portions of *HBG1* and *HBG2* genes. The sequence fragment with a single nucleotide difference between *HBG1* and *HBG2* genes is indicted by the asterisk, which overlap with the gRNA target sequence.

noise (S:B) ratio upon treatment with pomalidomide to move forward into high-throughput screening.

## Validation of HUDEP2_HBG1_HiBiT HbF expression

To further validate and confirm that the induced HiBiT signals correlate to the induction of HbF expression in HUDEP2_HBG1_HiBiT cells upon pomalidomide treatment, we employed three methods: 1) HbF positive cells (F-cells) analyzed by flow cytometric analysis, 2) *HBG1*-HiBiT luminescence signals measured by HiBiT protein complementation assay, and 3) protein expression levels of α-, β-, and γ-globin and HiBiT fusion proteins detected by western blot analysis. Three sets of cells were treated equally with pomalidomide at diluting

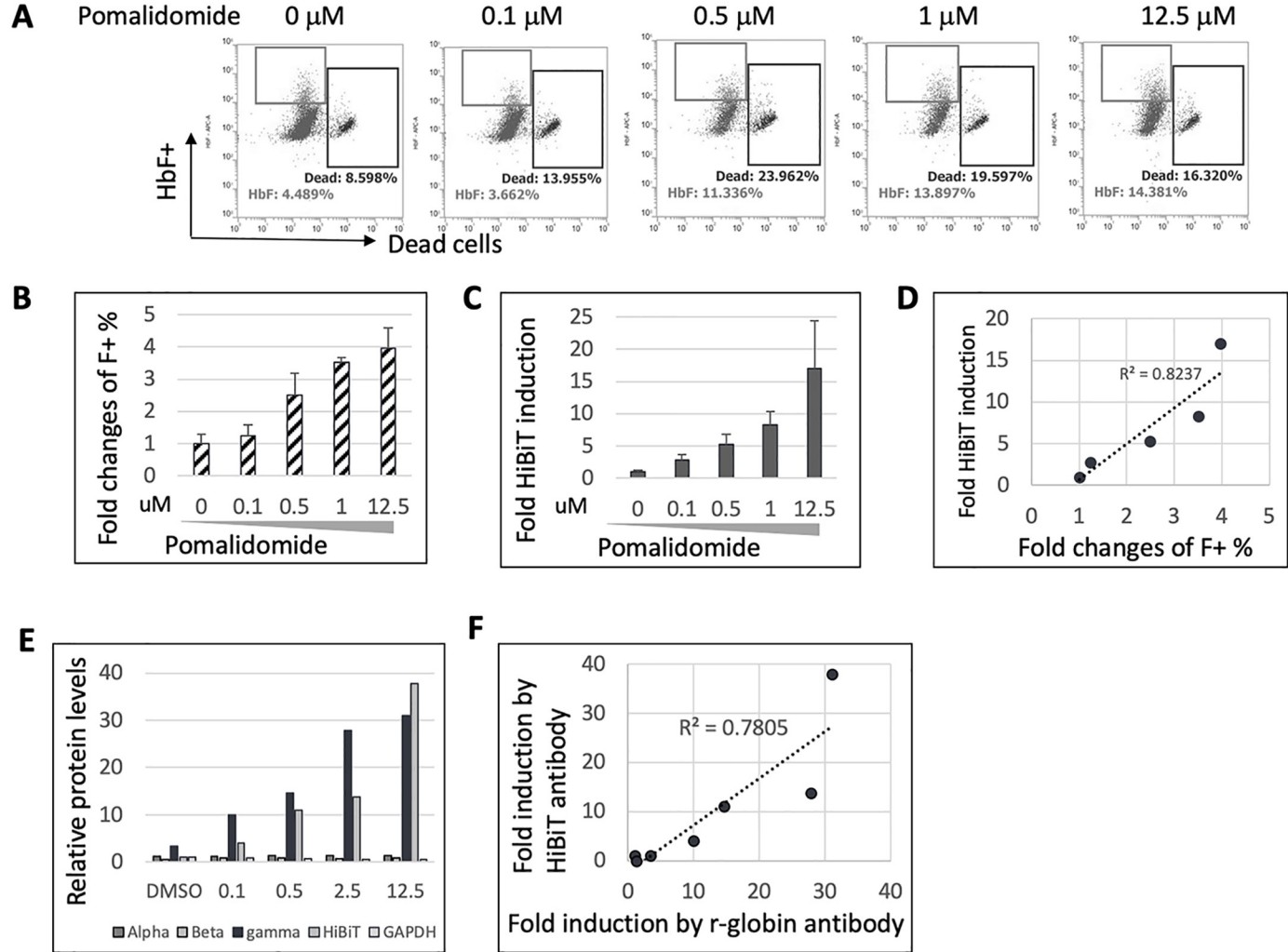

**Fig 2. Validation of HUDEP2_HBG1_HiBiT reporter cell line.** (A) Representative FACS results of HUDEP2_HBG1_HiBiT reporter cells treated with increasing concentrations of pomalidomide in differentiation media for 6 days. Gated populations of HbF positive cells and dead cells are indicated. (B) FACS data from (A) plotted as fold changes of percentage of HbF positive cells compared to DMSO as the negative control (n = 2). (C) Fold changes in HiBiT luminescence signals of HUDEP2_HBG1_HiBiT reporter cells treated with increasing concentrations of pomalidomide compared to DMSO as the negative control (n = 3). (D) Correlation of fold changes of HiBiT luminescence signals detected by Nano-Glo HiBiT lytic detection reagent and percentages of HbF positive cells detected by flow cytometric analysis. (E) Quantification of α, β, γ-globin proteins and HiBiT-tagged protein expression levels by western blot from S1 Fig. Fold changes of protein expression levels were calculated following normalization to GAPDH and then relative to DMSO treated samples. (F) Correlation of fold changes of HiBiT luminescence signals detected by Nano-Glo HiBiT Lytic Detection reagent and γ-globin protein expression levels detected by western blot analysis.

concentrations in differentiation media for 6 days. Pomalidomide increased the percentages of HbF positive cells detected by flow cytometric analysis (FACS) using anti-HbF antibody staining (Fig 2A and 2B and S1 File), as well as increased HiBiT luminescence signals detected by HiBiT luminescence assay (Fig 2C and S1 File), both in a dose-dependent manner. The fold changes of HbF-positive cell percentages by FACS and the fold changes of HiBiT luminescence signals were highly correlated ($R^2$ = 0.82), indicating the *HBG1*-HiBiT protein complementation assay is a robust surrogate readout for HbF positive percentage quantitated by FACS (Fig 2D).

We further validated the HUDEP2_HBG1_HiBiT reporter cell line by quantitating protein expression levels of α-, β-, and γ-globin and *HBG1*_HiBiT fusion protein by western blot

analysis. The molecular glue pomalidomide was chosen as a positive control drug due to its unique ability to reverse γ-globin transcriptional silencing [40]. Pomalidomide treatment significantly increased the protein expression levels of γ-globin and *HBG1*_HiBiT fusion proteins in a dose-dependent manner but had no measurable effect on the protein expression levels of α- and β-globin (Fig 2E and S1 Fig). The fold induction changes of protein expression levels of γ-globin and *HBG1*_HiBiT fusion proteins were highly correlated (Fig 2F), with a co-efficiency factor of 0.78, confirming that the HiBiT tag was successfully inserted into the C-terminus of the *HBG1* gene, and that HiBiT luminescence signals accurately reflect the expression levels of endogenous γ-globin protein.

## Optimization of a high-throughput-screen (HTS) erythroid HbF induction platform

To accommodate dense compound libraries for screening campaigns, we miniaturized the HiBiT protein complementation assay into 384-well format. Pomalidomide was employed as a positive control to optimize screening conditions (e.g., concentration of compound, cell seeding density, incubation time). First, pomalidomide at 10 mM in DMSO was spotted into 384-well plates using an acoustic dispensing system, then 50 μl of HUDEP2_HBG1_HiBiT cells at a concentration of either $2\times10^5$/ml or $1\times10^5$/ml in differentiation media were added to each well, yielding final drug concentrations of 10 μM and 1 μM, respectively. Cells were harvested for HiBiT luminescence detection after 4, 5 or 6 days of incubation with compound. Consistently, we observed induced HiBiT luminescence signals as early as 4 days incubation with pomalidomide at either concentration (Fig 3A). Luminescence signals improved with increased incubation time (Fig 3A–3C) for both drug treated and untreated samples. This resulted in modest fold change increases of HiBiT luminescence signals from both 1 μM and 10 μM pomalidomide-treated samples over DMSO controls from day 4 to day 6. Although there were no significant differences in S:B ratios between the different seeding densities, the variability of HiBiT luminescence signals from replicates was larger when half the number of cells were used (Fig 3B and 3D). The screening assay demonstrated an adequate dynamic range between positive (pomalidomide) and negative (DMSO) controls yielding a Z' factor of 0.5 with 5 days' incubation. We determined that $1\times10^4$ cells/well cell seeding density and a 5-day assay time in differentiation media were optimal conditions to move forward into high-throughput screening.

## Chemogenomic screen for HbF inducing compounds and targets

To pressure test the HUDEP2_HBG1_HiBiT reporter cell line and HbF induction conditions, we initiated a chemogenomic screening campaign to identify compounds capable of inducing fetal hemoglobin expression. Using annotated compound libraries (approximately 5000 clinical and FDA-approved compounds) from SelleckChem with known targets and/or mechanisms of action, we performed proof-of-concept screening with the optimized HTS workflow (Fig 4A). Briefly, compounds at 10 mM stocks were pre-spotted into 384-well assay plates, then HUDEP2_HBG1_HiBiT cells in differentiation media were added to each well to achieve final compound concentrations of either 1 μM or 10 μM. By screening at two concentrations, we aimed to increase the screen reproducibility. HiBiT luminescence signals were measured by adding Nano-Glo lytic detection reagent and read on an EnVison microplate reader after 5 days of compound exposure. Screening results of HiBiT luminescence signals from 1 μM and 10 μM compound screens are shown in Fig 4B and reveal good separation of HiBiT luminescence signals between negative (blue) and positive (red) controls. Hit compounds were defined as those inducing HiBiT luminescence signals above 3 standard deviations of the average

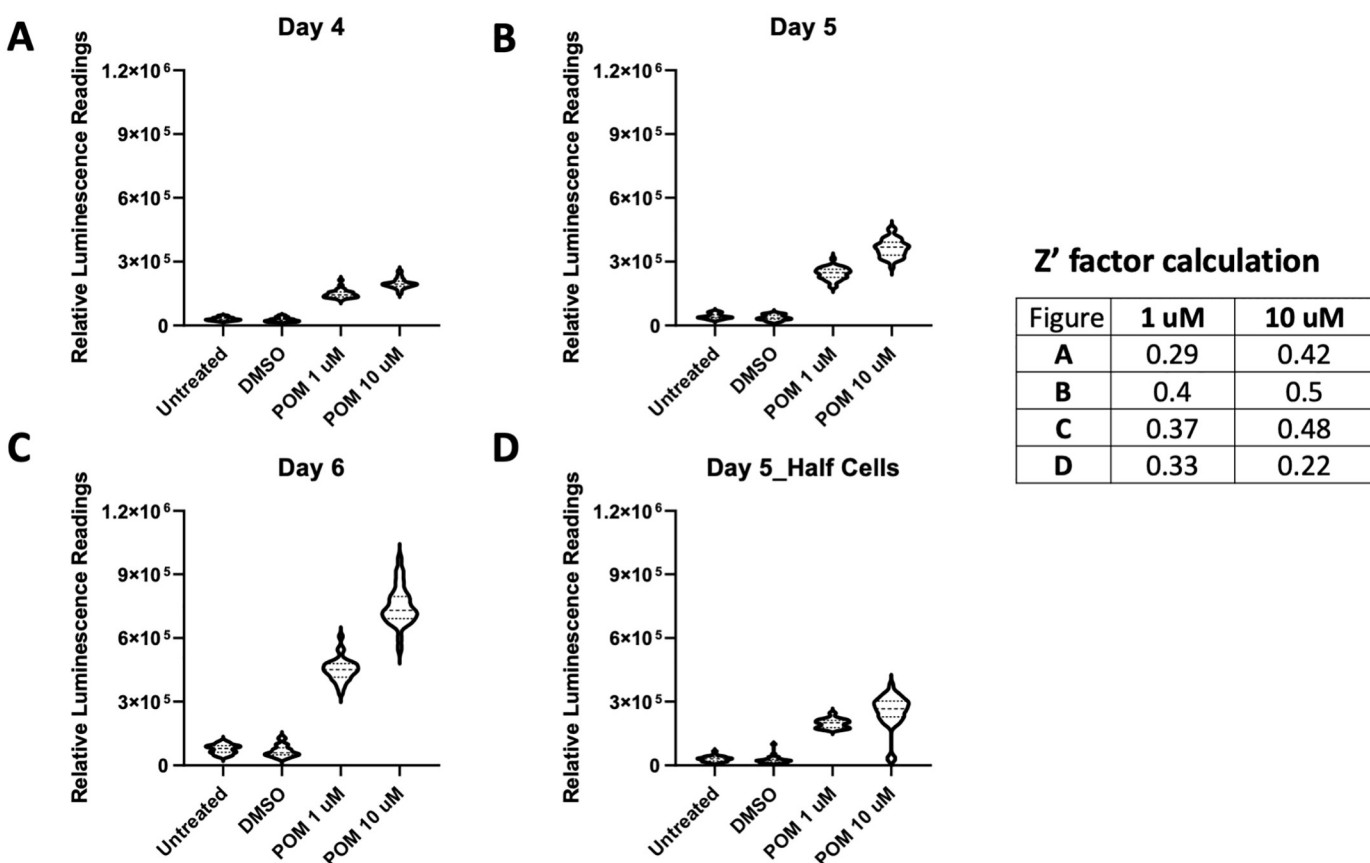

**Fig 3. Assay optimization for high-throughput screening.** Pomalidomide-treated HUDEP2_HBG1_HiBiT cells were harvested for HiBiT luminescence signal detection after 4 (A), 5 (B, D) or 6 (C) days of incubation. Fold changes of HiBiT luminescence signals compared to DMSO treatment samples are graphed. Average values of thirty-two samples for each treatment are presented. Z' factors were calculated using GraphPad Prism 8 software.

values of DMSO negative controls (Fig 4B, dotted lines). Using this hit cutoff, we identified 50 compounds from the primary chemogenomic screens and subsequently confirmed 10 hits that showed a 4-point dose response phenotype upon reconfirmation (Fig 4C, HiBiT assay and S2 File). Additionally, we included a CellTiter-Glo viability assessment during reconfirmation to determine if loss of HbF induction was due to cell toxicity (Fig 4C, CellTiter assay and S2 File).

As a side-by-side comparison with a clinical HbF inducer, we also tested FTX-6058 [30], an investigational EED-directed drug for SCD, in our high-throughput platform. We confirmed that FTX-6058 induced over 10-fold HiBiT luminescence signals at 1 μM treatment but demonstrated considerable toxicity at 3.3 μM and 10 μM treatments (Fig 4C).

### Validation of lead compound efficacy in primary erythroid CD34 + hematopoietic stem cells

We selected three compounds that demonstrated the highest HbF induction properties, pomalidomide, avadomide and idoxuridine, for further validation in human primary erythroid progenitor CD34+ cells. Both pomalidomide and avadomide treatments resulted in increased percentages of HbF positive cells at similar levels in a dose-dependent manner in CD34+ cells (Fig 5A and 5B and S3 File). Idoxuridine potently increased the percentage of HbF positive cells under 1 μM of drug treatment without noticeable toxicity.

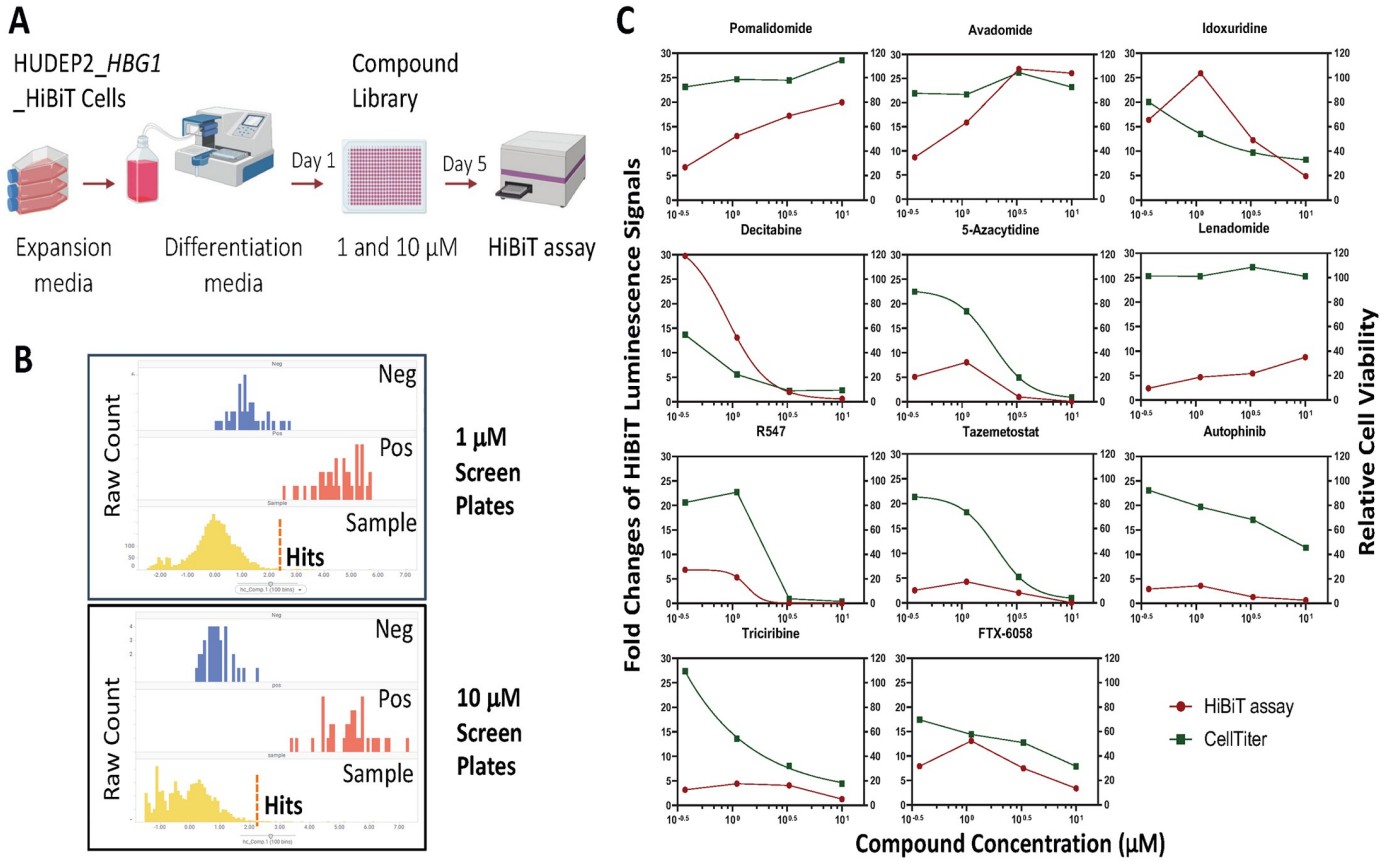

**Fig 4. High throughput chemogenomic screen for HbF inducers.** (A) Schematic diagram of HUDEP2_HBG1_HiBiT chemogenomic screen for identification of compounds and targets that up-regulate fetal hemoglobin gene expression. (B) Compound activity distribution and hit calling strategy. Primary hits were selected based on HiBiT luminescence signal +3xSTDEV above the average HiBiT luminescence signals of DMSO treated samples (dotted line). (C) HUDEP2_HBG1_HiBiT cells were treated with the indicated compounds at final concentrations of 0, 0.37, 1.1, 3.3 μM, and 10 μM for 5 days. HiBiT luminescence signals and relative cell viability were detected by Nano-Glo HiBiT Lytic Detection reagent and CellTiter Glo reagent, respectively. The data were normalized to DMSO controls for fold changes of HiBiT luminescence signals, and as 100% for the relative cell viability.

## Investigation of mechanisms of action of hit compounds

Numerous studies have reported that several transcription factors including BCL11A, IKZF1 and LRF/ZBTB7A suppress the expression of the γ-globin genes [31–33]. Extrapolating these data to our hit compounds, we next investigated whether pomalidomide, avadomide, and idoxuridine induce HbF expression via modulation of any of these transcriptional repressors. Parental HUDEP2 cells were treated with pomalidomide, avadomide or idoxuridine at various concentrations in differentiation media for 5 days, then cells were harvested to perform 1) FACS analysis of HbF positive cells using HbF antibody staining and 2) western blot analysis to detect protein expression levels in cell lysates. Consistent with the results from CD34+ cells that were treated with these compounds (Fig 5A and 5B), pomalidomide and avadomide increased the percentages of HbF positive cell populations in parental HUDEP2 cells in a dose-dependent manner, with idoxuridine again demonstrating the highest potency (Fig 6A). Interestingly, both pomalidomide and avadomide, but not idoxuridine, significantly downregulated BCL11A and IKZF1 protein levels in a dose-dependent manner (Fig 6B and 6C and S4 File). On the other hand, protein levels of LRF/ZBTB7A, and CRBN (a protein that forms an E3 ubiquitin ligase complex involved in protein degradation through binding with

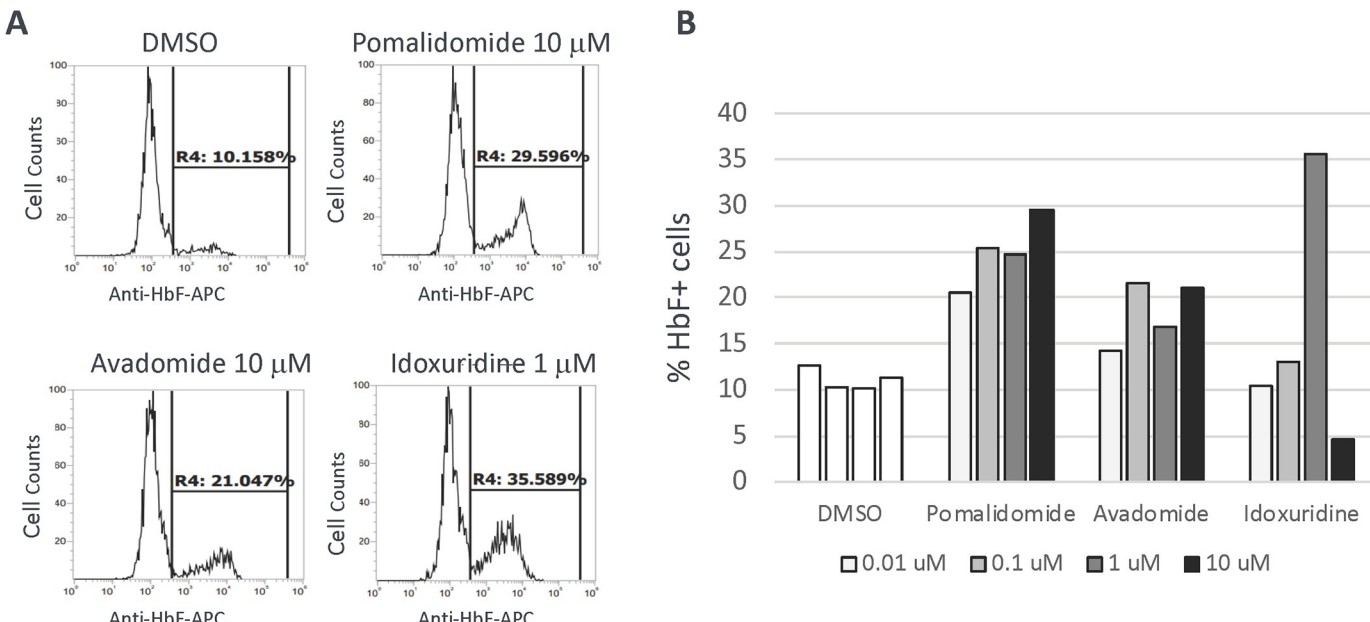

**Fig 5. Validation of the HbF inducing drugs in human primary CD34+ cells.** Human peripheral blood CD34+ cells treated with lead hit compounds and profiled by FACS for HbF expression. (A) Representative images of FACS analyses are shown. The percentages of fetal globin positive cells are indicated. (B) Percentages of HbF positive cells for each treatment are graphed.

thalidomide and its derivatives) [34] in HUDEP2 cells were modestly decreased by both pomalidomide and avadomide treatments but not by idoxuridine treatment (Fig 6B and 6C and S4 File), suggesting that the induction of HbF by these small molecules are via different mechanisms.

## Discussion

HbF induction has been proven to be an efficacious therapeutic approach in β-globin disorders such as β-thalassemia and SCD [35], as evidenced by recent encouraging reports from several clinical trials [36–39]. Although gene-based and cell-based therapies have shown potential to cure sickle cell disease, accessibility to such treatments is limited, particularly for patients in developing countries who make up the bulk of the SCD population. Pharmacological intervention to induce HbF remains challenging due to a dearth of therapeutic targets that can safely and efficaciously mediate the desired phenotype [24, 25]. Consequently, there are numerous efforts to find more effective and financially viable therapeutic classes of drugs with acceptable safety and tolerability profiles [24, 25].

To further the discovery of small molecule therapeutics to treat SCD, we developed an integrative workflow comprised of three components: 1) engineering of a robust human erythroid HbF reporter cell line; 2) high-throughput HbF phenotypic screening of dense small molecule libraries; and 3) orthogonal validation of lead HbF therapeutics in a preclinical primary human CD34+ HSC assay. The engineered HUDEP2_HBG1_HiBiT reporter cell line described here enables real time monitoring of endogenous γ-globin expression without modification of any regulatory elements that control hemoglobin expression. In comparison to conventional FACS-based compound screening assays, the HiBiT luminescence assay demonstrates higher sensitivity, considerably lower cost, and lower technical requirements.

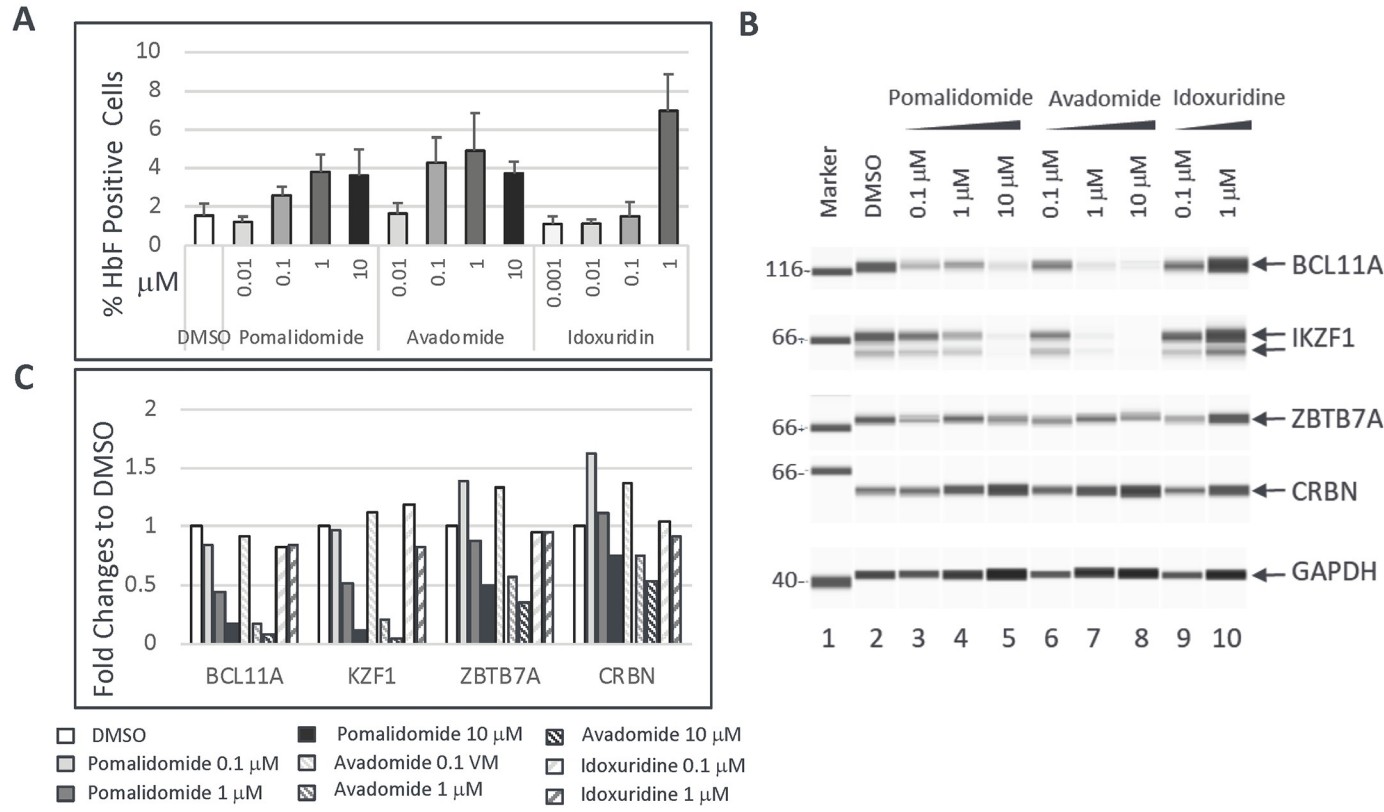

**Fig 6. Investigation of molecular targets of lead hit compounds.** Parental HUDEP2 cells were treated with the indicated compounds for 6 days. (A) Induction of HbF positive cells detected by FACS analysis. The average values and standard deviation of percentages of HbF positive cells from three replicates are graphed. (B) Western blot images of BCL11A, IKZF1, LRF/ZBTB7A, CRBN, and GAPDH proteins expression in DMSO- and compound-treated HUDEP2 cells. GAPDH was used as the loading control. (C) Quantification of BCL11A, IKZF1, LRF/ZBTB7A, and CRBN proteins levels from (B). Fold changes of protein expression levels were normalized with GAPDH controls and then fold change values determined relative to DMSO treated samples.

By profiling compound libraries consisting of clinical candidates and FDA-approved drugs, we identified 10 compounds that had HbF inductive activities in HUDEP2 cells. Among them, several reported HbF inducers such as pomalidomide [40–42], lenalidomide [35, 43], decitabine and 5-azacytidine [44], idoxuridine [45] and tazemetostat [46, 47] were identified, which added confidence and clinical relevance to our screening system. Importantly, we identified four novel HbF inducing small molecules, including avadomide, an analogue of thalidomide and pomalidomide [48]; triciribine, a DNA synthesis inhibitor that acts as a specific inhibitor of the Akt signaling pathway [49]; autophinib, a potent VPS34 inhibitor, selectively inhibiting starvation- and rapamycin-induced autophagy [50]; and R574, a potent ATP-competitive inhibitor of CDK1/2/4 [51]. None of these compounds have previously been reported to induce HbF expression, making this a first report of their therapeutic potential in SCD. Further extensive preclinical studies would be required to establish whether these drugs are suitable as potential treatments for SCD. Comprehensively, the integrative drug discovery strategy presented here enables novel target and pathway identifications as well as potential drug repositioning of clinically tested materials with new pharmacological mechanisms and therapeutic properties.

Thalidomide and its derivatives exert their therapeutic activity by targeting specific proteins to an E3 ubiquitin ligase for subsequent proteasomal degradation [52, 53]. Many zinc finger transcription factors such as BCL11A [40, 54], LRF/ZBTB7A [32], IKZF1 and IKZF3 [31, 55,

56] are affected by thalidomide derivatives, either directly via degradation or indirectly via the affected signaling pathways. Therefore, we investigated how these compounds affect the expression of several key transcriptional repressors of fetal globin gene expression, including BCL11A, IKZF1/Ikaros and LRF/ZBTB7A. We demonstrated in HUDEP2 cells that avadomide, similar to pomalidomide [40], but not idoxuridine, significantly downregulated the negative transcriptional regulators of HbF, such as BCL11A and IKZF1 in a dose-dependent manner, suggesting the HbF-activating properties of the degrader drugs are divergent from those of idoxuridine. Further studies to understand how these compounds affect transcriptional complexes and the mechanistic pathways in regulating HbF expression could lead to new therapeutic targets and modalities for selectively reactivating fetal hemoglobin in SCD patients.

In summary, we have developed an integrated cell-based phenotypic screening platform that enables dense high-throughput small molecule screening to identify HbF inducing compounds for hemoglobinopathy drug discovery. Application of the described screening platform to additional structurally diverse chemical libraries is the next phase in the discovery of HbF-inducing compounds, with the ultimate goal of developing novel therapeutics that outperform the clinical standard of care and offer improved health to SCD patients worldwide.

## Supporting information

**S1 Fig. Western blot images of α, β, γ-globin proteins and HBG1_HiBiT-tagged protein expression induced by pomalidomide treatment of HUDEP2_HBG1_HiBiT cells for 6 days.** GAPDH was used as the loading control. Lane 1 (P): parental HUDEP2 cells; Lane 2 (CM): HUDEP2_HBG1_HiBiT cells cultured in expansion media; Lane 3: HUDEP2_HBG1_HiBiT cells in differentiation media with 0.05% DMSO and without pomalidomide. Lane 4–7: HUDEP2_HBG1_HiBiT cells in differentiation media with various concentrations of pomalidomide.
(TIFF)

**S1 File. Supporting data for Fig 2.** Raw data from experiments assaying HUDEP_HBG1_HiBiT reporter cells for viability (CellTiter), HiBiT expression and %HbF (F+) expression by FACS in the presence of increasing concentrations of pomalidomide.
(XLSX)

**S2 File. Supporting data for Fig 4.** Raw data from experiments assaying HUDEP_HBG1_HiBiT reporter cells for viability (CellTiter) and HiBiT expression in the presence of increasing concentrations of lead hit drugs.
(XLSX)

**S3 File. Supporting data for Fig 5.** Raw data from FACS experiments assaying human primary CD34+ cells for HbF expression in the presence of lead hit drugs.
(XLSX)

**S4 File. Supporting data for Fig 6.** Raw data from western blot experiments assaying HUDEP_HBG1_HiBiT reporter cells for expression of key transcription factors and mechanistic proteins in the presence of lead hit drugs.
(XLSX)

## Acknowledgments

We thank the members of Core Biology at Takeda Development Center Americas, Inc. for their meaningful discussions and technical support. Human umbilical cord blood-derived erythroid progenitor (HUDEP-2) cell line was obtained from RIKEN Institute, Japan.

## Author Contributions

**Conceptualization:** Jian-Ping Yang, Sarah J. Fink, Sebastien Vallee, Jon Kenniston, Nikolaos Papaioannou, Narender R. Gavva, Shane R. Horman.

**Data curation:** Matt Petrus.

**Formal analysis:** Sebastien Vallee.

**Funding acquisition:** Jon Kenniston, Nikolaos Papaioannou.

**Investigation:** Jian-Ping Yang, Rachel Toughiri, Anshu P. Gounder, Matt Petrus.

**Methodology:** Jian-Ping Yang, Rachel Toughiri, Anshu P. Gounder, Matt Petrus.

**Project administration:** Jian-Ping Yang, Jon Kenniston, Shane R. Horman.

**Resources:** Dan Scheibe, Shane R. Horman.

**Software:** Dan Scheibe, Matt Petrus.

**Supervision:** Jian-Ping Yang, Sarah J. Fink, Sebastien Vallee, Jon Kenniston, Nikolaos Papaioannou, Steve Langston, Narender R. Gavva, Shane R. Horman.

**Writing – original draft:** Jian-Ping Yang, Shane R. Horman.

**Writing – review & editing:** Jian-Ping Yang, Sarah J. Fink, Sebastien Vallee, Jon Kenniston, Steve Langston, Narender R. Gavva, Shane R. Horman.

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
