## [Decision Letter · Decision Letter 0]

2 Aug 2024

PONE-D-24-26300Identification of small molecule agonists of fetal hemoglobin expression for the treatment of sickle cell diseasePLOS ONE

Dear Dr. Horman,

Thank you for submitting your manuscript to PLOS ONE. After careful consideration, we feel that it has merit but does not fully meet PLOS ONE’s publication criteria as it currently stands. Therefore, we invite you to submit a revised version of the manuscript that addresses the points raised during the review process.

**The reviewers see great value in this study and the potential to improve therapies to augment hemoglobin F% in patients with sickle cell disease.  There are only minor comments that the reviewers have requested to be addressed, such as the rationale to use pomalidomide and editing issues.**

We look forward to receiving your revised manuscript.

Kind regards,

Santosh L. Saraf

Academic Editor

PLOS ONE

Journal Requirements:

Takeda Development Center Americas, Inc. provided sponsorship and financial support for this study. All the authors are employees of Takeda Pharmaceutical Industries, Ltd., and had equity ownership with Takeda Pharmaceutical Industries, Ltd.

We note that one or more of the authors are employed by a commercial company: Takeda Pharmaceutical Industries, Ltd. 

“The funder provided support in the form of salaries for authors, but did not have any additional role in the study design, data collection and analysis, decision to publish, or preparation of the manuscript. The specific roles of these authors are articulated in the ‘author contributions’ section.”

The full raw screening data can be provided as a supplementary file if the editors require.

Reviewers' comments:

Reviewer's Responses to Questions

**Comments to the Author**

1. Is the manuscript technically sound, and do the data support the conclusions?

Reviewer #1: Yes

Reviewer #2: Yes

Reviewer #3: Yes

2. Has the statistical analysis been performed appropriately and rigorously? 

Reviewer #1: Yes

Reviewer #2: Yes

Reviewer #3: N/A

3. Have the authors made all data underlying the findings in their manuscript fully available?

Reviewer #1: Yes

Reviewer #2: Yes

Reviewer #3: Yes

4. Is the manuscript presented in an intelligible fashion and written in standard English?

Reviewer #1: Yes

Reviewer #2: Yes

Reviewer #3: Yes

5. Review Comments to the Author

**Reviewer #1:** This manuscript describes a new screening system for HbF-inducing drugs. There is a great need for these drugs because gene therapy and BM transplantation can only be used to treat a relatively small number of SCD and beta-thal patients. Four new HbF-inducers were identified. It will be of great interest to determine whether these drugs are active in vivo in SCD mice and nonhuman primate studies.

**Reviewer #2: **A novel approach to addressing the issue of discovering new and/or existing therapeutics to induce fetal hemoglobin as a treatment option for individuals with sickle cell disease. Methods are clearly stated and easily reproducible/replicated—utilization of CRISPR and modern methods to accomplish goals laid out in the manuscript.

Minor Comments

- Line 257: add brief sentence about why pom was chosen as positive control over other agents

- Please proofread again; minor errors were found

o Line 66 misspelling of azacitidine

o Line 196 incorrect capitalization use in the word “complete”

**Reviewer #3:** hydoxyurea is the only drug approved for sickle cell disease which induces fetal hemoglobin. The author and colleagues identified small molecules that also induce fetal hemoglobin in cell lines. This is a study that is needed and is valuable and we need more drugs that can be used to help our patients with sickle cell disease.

6. PLOS authors have the option to publish the peer review history of their article (what does this mean?). If published, this will include your full peer review and any attached files.

Reviewer #1: No

Reviewer #2: No

Reviewer #3: No

---

## [Author Response · Author response to Decision Letter 0]

27 Sep 2024

The manuscript has been reformatted to PLOS One formatting standards. Additionally, we have updated the Competing Interests and Funding Statement in both the manuscript and cover letter. All figures have been provided individually in .TIF format and adhere to the journal's requirements.We have also included all of the data that was used to produce the data figures and have included those files as "Supporting Data" files. We have provided all of the original uncropped western blot images for the western blot in fig 6. Reviewers 1 and 3 did not have any suggested edits. Reviewer 2 submitted the following suggestions and we have made those changes:

Reviewer 2 Minor Comments

- Line 257: add brief sentence about why pom was chosen as positive control over other agents

- Please proofread again; minor errors were found

o Line 66 misspelling of azacitidine

o Line 196 incorrect capitalization use in the word “complete”

---

## [Editor Report · Decision Letter 1]

4 Oct 2024

Identification of small molecule agonists of fetal hemoglobin expression for the treatment of sickle cell disease

PONE-D-24-26300R1

Dear Dr. Horman,

We’re pleased to inform you that your manuscript has been judged scientifically suitable for publication and will be formally accepted for publication once it meets all outstanding technical requirements.

Kind regards,

Santosh L. Saraf

Academic Editor

PLOS ONE

---

## [Editor Report · Acceptance letter]

28 Oct 2024

PONE-D-24-26300R1 

PLOS ONE

Dear Dr. Horman, 

I'm pleased to inform you that your manuscript has been deemed suitable for publication in PLOS ONE. Congratulations! Your manuscript is now being handed over to our production team.

Kind regards, 

on behalf of

Dr. Santosh L. Saraf 

Academic Editor

PLOS ONE